# Optimization of a Microencapsulation Process Using Oil-in-Water (O/W) Emulsion to Increase Thermal Stability of Sulforaphane

**DOI:** 10.3390/foods12203869

**Published:** 2023-10-22

**Authors:** Víctor Zambrano, Rubén Bustos, Yipsy Arozarena, Andrea Mahn

**Affiliations:** 1Department of Chemical Engineering, University of Santiago of Chile, Avenida Libertador Bernardo O’Higgins 3363, Estación Central, Santiago 9170019, Chile; victor.zambrano@usach.cl (V.Z.); ruben.bustos@usach.cl (R.B.); 2Food Science and Technology Doctorate Program, University of Santiago of Chile, Avenida Libertador Bernardo O’Higgins 3363, Estación Central, Santiago 9170019, Chile

**Keywords:** sulforaphane, thermal stability, entrapment efficiency, oil-in-water emulsion

## Abstract

Sulforaphane (SFN) is a bioactive compound widely studied for its potential applications in pharmaceutical, nutraceutical, and food industries since it offers health benefits due to its nature as a Phase 2 enzyme inducer. Its application in the food industry has been limited because SFN is unstable at high temperatures in an aqueous milieu. An option to increase SFN stability and protect it from thermal degradation is microencapsulation. The aim of this work was to optimize a microencapsulation process using oil-in-water emulsion to increase the thermal stability of SFN. The operation conditions that gave the highest entrapment efficiency were determined via experimental design and response surface methodology. Thermal degradation of microencapsulated SFN was studied at 37, 50, 60, and 70 °C. The optimum microencapsulation conditions were 8 min stirring, SFN/Gum Arabic ratio of 0.82, and surfactant/oil ratio of 1.0, resulting in an entrapment efficiency of 65%, which is the highest reported so far. The thermal stability of microencapsulated SFN was greatly enhanced compared with free SFN, with a 6-fold decrease in the degradation kinetic constant and a 41% increase in the activation energy. These results will contribute to a more efficient incorporation of SFN in various food matrices and explore new microencapsulation technologies to maximize the efficiency and stability of SFN.

## 1. Introduction

Sulforaphane (SFN) is an isothiocyanate that comes from glucoraphanin (GFN), the most abundant glucosinolate found in broccoli. The health-promoting effects of SFN are largely documented, and they include the prevention and treatment of several types of cancer and protection against cardiac diseases, hypertension, and neurodegenerative diseases, among others [1,2]. This is associated with its capacity to induce detoxifying Phase 2 enzymes [3].

In order to exploit the health benefits of SFN, different strategies have been proposed to incorporate it in food matrices based on processing *Brassicaceae* vegetables such as broccoli and cabbage as natural sources of SFN. Drying [4], blanching [5], and fermentation [6] are some of the processes studied to increase SFN content and stabilize it in food matrices. However, further development and industrialization of those technologies have been limited due to the thermal instability of SFN. Accordingly, SFN stabilization remains a technological challenge. In this regard, microencapsulation appears as an attractive technology to stabilize SFN after its extraction from a vegetal source. SFN is poorly water-soluble [7]; therefore, microencapsulation seems adequate because it is intended to stabilize hydrophobic compounds and isolate them from a degradation-promoting environment [8,9].

Wu et al. [10] studied the microencapsulation of SFN using simple and complex coacervation followed by spray drying and evaluated the stability of the powder during storage at 35 °C. The highest stabilization was achieved using maltodextrin as wall material and 170 °C air temperature at the inlet of the spray dryer. In these conditions, the degradation kinetics constant was 0.0361/d, one order of magnitude lower than free SFN. Fahey et al. [11] obtained a similar result by using an inclusion complex with cyclodextrins followed by freeze-drying. Both authors studied the stability of dehydrated SFN microcapsules, and then these results cannot be extrapolated to humid food matrices. 

Wu et al. [12] investigated the thermal degradation kinetics of SFN in ethyl acetate broccoli extract and free SFN in aqueous solutions at different pHs in the temperature range from 60 to 90 °C. They found that SFN stability is higher at acidic pH and slightly higher in the broccoli extract in comparison with free SFN, with around a 1.8-fold reduction in the degradation kinetics constant at 90 °C. This suggests that incorporating SFN into acidic food matrices would contribute to protecting SFN from thermal degradation. Martínez-Hernández et al. [13] studied the thermal stability of SFN microencapsulated with cyclodextrins, reporting no significant stabilization of SFN, unlike for GFN, which was retained up to 24 h at 90 °C, using relatively high amounts of cyclodextrin (90 mmol/L). These cyclodextrin concentrations are not applicable in the food industry due to their high price. 

In addition to increasing the thermal stability of SFN, microencapsulation entrapment efficiency is a relevant parameter in terms of industrial processing. Simple and complex coacervation, as well as inclusion complex, show low entrapment efficiencies of SFN, ranging from 12% to 39% [10,14]. Higher entrapment efficiencies were attained by nanoencapsulation. Danafar et al. [15] obtained around 86% entrapment efficiency using mPEG-PCL nanoparticles followed by lyophilization. Manjili et al. [16] obtained 87% efficiency using iron oxide nanoparticles, and Soni et al. [17] achieved 85% efficiency using nanostructured lipid carriers. No information about the thermal stabilizing effect of these nanoencapsulation techniques was reported. In those studies, the nanoencapsulation systems were intended for pharmaceutical applications, and FDA does not recognize the encapsulating materials used as safe for food. Additionally, the encapsulation materials used are highly expensive.

Currently, there is no information about SFN microencapsulation using oil-in-water systems, nor information about the thermal stabilization of SFN via microencapsulation methods scalable to the food industry that include degradation kinetic studies and entrapment efficiency. The microencapsulation of isothiocyanates other than SFN was approached. Li et al. [18] microencapsulated benzyl isothiocyanate via an inclusion complex using β-cyclodextrin, obtaining an 84% entrapment efficiency. Ko et al. [19] microencapsulated allyl isothiocyanate by emulsification using gum Arabic as wall material, with an entrapment efficiency of 85%. There is no information about the thermal degradation kinetics of these microencapsulated isothiocyanates. 

Microencapsulation of SFN via oil-in-water emulsification seems promising to improve the thermal stability of the compound in food matrices, given its amphiphilic properties [20]. Zhang et al. [21] reported that the addition of oil to broccoli suspensions increased the SFN stability of the dried powder at 25 °C. Li et al. [18] microencapsulated allyl isothiocyanate using oil-in-water nanoemulsion and attained 78% remaining after 60 days at 30 °C in aqueous solution. There are no studies on the microencapsulation of SFN using oil-in-water emulsification, and there is no information about the thermal stabilizing effect of such a method on isothiocyanates. 

Given that emulsification offers high entrapment efficiencies and that oil protects isothiocyanates from thermal degradation in an aqueous environment, we propose that using an oil-in-water emulsification system to microencapsulate SFN will result in higher entrapment efficiency and will improve the thermal stability of SFN in aqueous solution. The aim of this work was to investigate the microencapsulation conditions using oil-in-water emulsion that maximizes entrapment efficiency and to study the effect of microencapsulation on the thermal stability of SFN. 

## 2. Materials and Methods

### 2.1. Materials

Broccoli seeds were provided by a single local supplier (Santiago, Chile). Deionized water was used throughout the study. Dichloromethane (Sigma–Aldrich, St Louis, MO, USA) and acetonitrile (Merck, Darmstadt, Germany) were HPLC grade. Ethanol (Merck, Darmstadt, Germany), Gum Arabic (GA) (Sigma–Aldrich, St Louis, MO, USA), Vaseline (Michelson, Santiago, Chile), anhydrous sodium sulfate (Winkler, Santiago, Chile), and Tween 80 (Merck, Darmstadt, Germany) were used. Sulforaphane standard was purchased from Sigma-Aldrich (Schnelldorf, Germany).

### 2.2. Sulforaphane Extraction

SFN extraction was performed according to the method proposed by Campas-Baypoli et al. [22] with some modifications. For maximum conversion of glucoraphanin to SFN, broccoli seeds (15 g) were ground and homogenized in 24 mL of deionized water and incubated at 45 ± 1 °C in a RE300 water bath (Stuart, Staffordshire, UK). After incubation for 3 h, the mixture was combined with 13 g anhydrous sodium sulfate and extracted twice with 100 mL of dichloromethane in a sonication bath (Sonics & Materials, Inc., Model VCX750/VCX500, Newtown, CT, USA) for 0.5 h. The organic extracts were vacuum filtered, pooled, and evaporated in a rotary evaporator (EV400H-V, Labtech RS.r.L., Sorisole, Italy) at 30 °C until complete removal of the organic solvent. Finally, the oily residue was recovered in 99.6% ethanol (analytic grade).

### 2.3. Experimental Designs

Preliminary assays were conducted aiming at defining the composition of the microencapsulation system: type of surfactant, type of oil phase, gum Arabic concentration, and oil/aqueous phase ratio that gives the highest stability of the emulsion system. The responses were the time before creaming begins (min) and the time needed to complete phase separation (min). Oil/aqueous phase ratio and gum Arabic concentration were studied via a factorial 4^1^ × 2^1^ design whose response was phase separation time, as shown in Table 1.

Optimization of the microencapsulation process was performed using a Box–Behnken design with three central points. The experimental factors were homogenization time (2, 5, and 8 min), SFN/GA mass ratio (0.4, 0.7, and 1.0 µg SFN/mg GA), and surfactant-to-oil mass ratio (SOR; 0.2, 0.6, and 1.0 g Tween80/g oil phase). The levels in this design were chosen based on preliminary microencapsulation experiments. The response variable was entrapment efficiency (%). Table 2 shows the experimental design and the responses.

### 2.4. Microencapsulation of SFN

The aqueous phase consisted of a 20% (w/w) GA solution with a pre-weighed quantity of Tween80, prepared by stirring at 600 rpm and 50 °C for one h. The oil phase consisted of dried SFN extract resuspended in 99.6% ethanol combined with Vaseline in 1:1 v/v proportion, mixed using a vortex for 30 s. Emulsions were prepared by adding the oil phase to the GA-Tween80 solution, homogenized with a blade homogenizer (Ultra-turrax TR 50, IKA-Werke GmbH, Staufen, Germany) using a rotational speed of 430× *g*. The emulsions were dried at 37 °C in a vacuum oven (Cole & Palmer, model 60061 Holdpack, Vernon Hills, IL, USA) at −0.6 MPa for 36 h. The dried microcapsules were pulverized in a porcelain mortar. The entrapment efficiency (*EE*) was calculated as the ratio of SFN mass (µg) in the microcapsules and the SFN mass (µg) added to the microencapsulation mixture, according to Equation (1), where EE is entrapment efficiency (%).
(1)$$EE\;(\%)=\frac{\mu g\;SFN\;content\;in\;the\;final\;microcapsules}{\mu g\;SFN\;added\;for\;microencapsulation}\times 100$$

### 2.5. Characterization of Microcapsules

The SFN microcapsules obtained in the optimal process conditions were characterized in terms of size, microstructure, color, and sulforaphane content.

#### 2.5.1. Particle Size and Microstructure

The emulsions and the rehydrated SFN microcapsules were observed in an optical microscope (Euromex Scope series 100–400 V, Arnhem, The Netherlands) equipped with a digital camera. A 100× magnification was used. Images were acquired with the software ImageFocus Alpha for Windows v1.3.7.22259 (Euromex Microscope BV, Arnhem, The Netherlands). An amount of 350 particles was randomly selected, each diameter was recorded, and the average particle diameter was reported. The microstructure of the dehydrated SFN microcapsules was analyzed by Scanning Electron Microscopy (SEM). SEM images of the microcapsules were taken on gold palladium-coated samples using a TESCAN (Brno, Czech Republic) Model VEGA 3 Electron microscope.

#### 2.5.2. Sulforaphane Content

The sulforaphane content was assessed by reverse phase HPLC, using the method proposed in previous works [23] with some modifications. For SFN extract, 1 mL of organic extracts vacuum filtered was evaporated in a rotary evaporator at 30 °C. The residue was dissolved in 2 mL acetonitrile and then filtered through a 0.22 µm membrane filter prior to injection into HPLC. The equipment was an HPLC-DAD Shimadzu (Tokyo, Japan), and a reversed-phase C18 column (5 µm, 250 × 4.6 mm particle size) (Agilent Technologies, Santa Clara, CA, USA) was used. The solvent system consisted of 20% acetonitrile in water; this solution was then changed linearly over 10 min to 60% acetonitrile and maintained at 100% acetonitrile for 5 min to purge the column. The column oven temperature was set at 30 °C. The flow rate was 1 mL/min, and 20 µL aliquots were injected into the column. Sulforaphane was detected by absorbance at 254 nm. Quantification was carried out by comparison with a sulforaphane standard curve. To build this curve, the standard was dissolved in acetonitrile, and eight injections of different volumes were made. Finally, the peak areas were plotted against the standard mass (0.056–6.75 µg). The linearity of the standard curve was expressed in terms of the determination coefficient (R^2^ = 0.99).

SFN content in microcapsules was assessed using the method proposed in [14] with some modifications. An amount of 300 mg of the powder was weighed, and 5 mL of dichloromethane was added, sonicated for 15 min, and reposed for 1 h. The extract was filtered through a filter paper, the solvent was evaporated in a rotary evaporator at 30 °C and reconstituted in 2 mL of acetonitrile. Sulforaphane was quantified by reverse-phase HPLC as described above.

#### 2.5.3. Color Analysis

Digital images of microcapsule powder were taken using a computer vision system with a digital camera (Canon Inc., Model EOS1000D, Taiwan). Parameters of the camera were calibrated using 30 color charts with a Minolta colorimeter, and the red, green, and blue color values (RGB) were obtained using the Adobe Photoshop CS3 software v10.0.1 (Adobe Systems Incorporated, 2007). The RGB values were converted to CIEL*a*b* space using the colorimetric converter of the colorizer website (https://colorizer.org/ accessed on 7 October 2023).

#### 2.5.4. Fourier Transform Infrared (FTIR) Spectroscopy Analysis

Sulforaphane microcapsules were characterized by FTIR spectroscopy with attenuated total reflectance (ATR) utilizing Perkin Elmer Spectrum 2 equipment (Llantrisant, UK). The bands were analyzed within a frequency range between 400 and 4000 cm^−1^, and the microcapsule wall polymer gum Arabic was used as a reference.

### 2.6. Thermal Stability Study

The stability of microencapsulated SFN was investigated at 37 °C, 50 °C, 60 °C, and 70 °C using a thermostatic water bath. Dried microcapsules (1 g) were resuspended in 5 mL HPLC-grade water in 20 mL vials wrapped with aluminum foil and stored at different temperatures for 15 h (50 °C, 60 °C, and 70 °C) or 27 h (37 °C). The controls consisted of free SFN extract resuspended in 5 mL HPLC-grade water. Five samples were taken in each condition at regular time intervals until completing the predefined time. The SFN concentration in the microcapsules was determined at each experimental point and reported as a fraction of the initial SFN content in the microcapsules (*C/C*_0_).

SFN degradation was described via a first-order kinetic model [24], as shown in Equation (2), where k is the rate constant, *t* is time, and *C*_0_ and *C* are the SFN concentration at *t* = 0 and *t* = t, respectively.
(2)$$\ln\frac{C}{C_{0}}=kt$$

### 2.7. Estimation of Activation Energy (Ea)

It is accepted that the kinetic constant depends on temperature following an Arrhenius equation, as shown in Equation (3). Here, *k*_0_ is the rate constant at a reference temperature (*T_ref_*), *T* is temperature, *Ea* is the activation energy, and *R* is the universal gas constant.
(3)$$\ln\frac{k}{k_{0}}=-\frac{E_{a}}{R}\cdot (\frac{1}{T_{ref}}-\frac{1}{T})$$

Degradation kinetic constants at each temperature were estimated by adjusting Equation (2) to the experimental data. The activation energy for SFN degradation was obtained from an adjustment of Equation (3) to the estimated kinetic constants obtained at temperatures above 40 °C, the temperature at which SFN degradation begins [24]. 

### 2.8. Statistical Analysis

Experimental design and statistical analyses were performed with Statgraphics Centurion XV. An analysis of variance (ANOVA) test was completed to evaluate the significance of the effects (*p* < 0.05). The optimization design was analyzed by response surface methodology, using a polynomial model to describe the experimental behavior (Equation (4)).
(4)$$y=b_{0}+b_{1}x_{1}+b_{2}x_{2}+b_{3}x_{3}+b_{4}x_{1}x_{2}+b_{5}x_{1}x_{3}+b_{6}x_{2}x_{3}+b_{7}x_{1}^{2}+b_{8}x_{2}^{2}+b_{9}x_{3}^{2}$$

The coefficients of the polynomial were represented by *b*_0_ (constant term), *b*_1_, *b*_2,_ and *b*_3_ (linear effect), *b*_7_, *b*_8,_ and *b*_9_ (quadratic effect), and *b*_4_, *b*_5,_ and *b*_6_ (interaction effects). The model quality was assessed by the determination coefficient (R^2^), and the optimum microencapsulation conditions predicted by the regression model were validated experimentally.

## 3. Results and Discussion

### 3.1. Selection of the Microencapsulation System

In the first stage, the effect of the oil/aqueous phase ratio and the concentration of gum Arabic in the aqueous phase on the stability of the emulsion was studied. Table 1 shows the experimental conditions and the responses. The oil/aqueous phase ratio and gum Arabic concentration in the aqueous phase significantly affected emulsion stability measured as initial and total phase separation time (*p* < 0.05). The most stable emulsion was obtained using a 1:1 oil/aqueous phase ratio and 20% (w/w) gun Arabic in the solution, remaining stable for more than 4 h (run A4). Accordingly, these conditions were used in the next stages.

### 3.2. Optimization of SFN Entrapment Efficiency 

A Box–Behnken design was used to identify the microencapsulation conditions that maximize SFN entrapment efficiency. Table 2 shows the experimental conditions and the responses. The lowest EE came from runs 7 and 12 (23 ± 1 and 22 ± 2%, respectively), using the lowest SOR in both cases. The highest EE was obtained in runs 2 and 3 (63 ± 3 and 65 ± 4%, respectively), using the highest SOR in both runs. The effect of the experimental factors on EE is given in Figure 1. SOR and SFN/gum Arabic mass ratio had significant positive effects on EE (*p*-values of 0.000 and 0.0015, respectively); i.e., an increase in SOR or in SFN/GA produces an increase in EE. Stirring time had no significant effect on the response, probably because of the high rotational speed used in all the experiments (8000 rpm) that could mask the effect of stirring time. In addition, the binary interactions that involve the significant factors had a statistically significant effect on EE, with the interaction of SFN/GA with itself showing the only negative effect (*p* = 0.0016). Equation (5) shows the regression model that describes the system considering only the significant factors. The determination coefficient (R^2^) was 85%; thus, Equation (5) explains 85% of the variability of the response. Here, EE is the entrapment efficiency, SOR is the surfactant-to-oil phase ratio, t is the stirring time, and SFN/GA is the SFN/gum Arabic mass ratio. The response surfaces obtained with the regression model (Equation (5)) are presented in Figure 2. Figure 2A suggests that there is an SFN/GA ratio (around 0.7) that maximizes EE. It also comes out that as SOR increases, as does EE, probably because of the stabilizing role of the surfactant. SOR showed no optimum. Figure 2C suggests that stirring time is directly proportional to EE, but since this factor had no significant effect, it cannot be optimized.
(5)$$EE=40.79-14.08\cdot \left(\frac{SFN}{GA}\right)+4.08\cdot \left(SOR\right)+4.53\cdot t^{2}+3.5\cdot t\cdot \left(SOR\right)+4.0\cdot \left(SOR\right)\cdot (SFN/GA)-5.81\cdot ({SOR)}^{2}$$

Equation (5) was used to determine the optimal microencapsulation conditions that result in the highest EE, yielding SOR = 0.62; SFN/GA = 1. Since stirring time had no significant effect, we chose to use the highest level (8 min) to obtain smaller particles. In these conditions, the theoretical EE was 65.1%. A run using the optimum conditions was executed in triplicate, and the resulting EE was 64.1 ± 1.2%. This EE is higher than the previously reported for microencapsulation systems using food-grade wall materials, as shown in Table 3. Complex coacervation using gelatine–gum Arabic and gelatine–pectin systems gave EE of 12.17 ± 0.10% and 17.91 ± 1.27%, respectively [14]. Microencapsulation of SFN via spray drying using maltodextrin, gum Arabic, and k-carrageenan as wall materials resulted in EE between 12 and 40% [10]. Complex coacervation using maltodextrin–gum Arabic and β-cyclodextrin–gum Arabic gave EE within the same range [12]. SFN-loaded broccoli membrane vesicles yielded an EE of 28.16 ± 5.05% [25]. The authors dissolved SFN in the aqueous phase, which could probably reduce EE given the low water-solubility of SFN. In the present work, SFN was dissolved in the organic phase, which partly explains the higher EE attained.

Other works developed SFN nanoparticles and micelles using mineralized hyaluronic acid–Stetradecyl nanocarriers [26], polymeric nanocarriers loaded with superparamagnetic iron oxide nanoparticles [27], nanostructured lipid carriers [17], and prolamin-based composite nanoparticles [28]. Nanoencapsulation and micelles methods resulted in EE up to 93%, but they are not likely to be used in the food industry because they represent a significantly higher production cost in comparison with microencapsulation methods.
foods-12-03869-t003_Table 3Table 3Micro- and nanoencapsulation methods used for SFN stabilization. NI means “not informed”.Wall MaterialEncapsulation MethodDrying MethodEntrapment Efficiency (%)ReferencesMicroencapsulation α-CDInclusion complexLyophilization NI[11]HP-β-CDInclusion complexRota evaporationNI[29]Gelatin/GAComplex coacervationVacuum oven 12.2 ± 0.1 [14]Gelatin/pectinComplex coacervationVacuum oven17.9 ± 1.3[14]GA/β-CD-Spray drying29.4 ± 2.9[12]Maltodextrin/GA-Spray drying34.0 ± 3.5[12]GA-Spray drying39.8 ± 1.5[12]Nanoencapsulation Broccoli membrane vesiclesNanoencapsulation -41.56 ± 8.56[25]Fe_3_O_4_ nanoparticles coated with gold PEGilated gold [(PEGylated Fe_3_O_4_@Au) NPsNanoencapsulation-65 ± 0.1[16]Copolymer (etilenoglycol)-poli (ε-caprolactone) (mPEG-PCL) nanoparticlesMicelles Lyophilization86.0 ± 1.6[15]Nanostructured lipids transporters—PEGUltrasound fusion emulsificationLyophilization84.9 ± 3.8[17]PCL-PEG-PCL nanoparticlesNanoencapsulationLyophilization87.1 ± 1.6[16]polymeric nanocarriers loaded with superparamagnetic iron oxide nanoparticlesNanoencapsulation-90.3 ± 0.4[27]Mineralized hyaluronic acid-Stetradecyl nanocarriersNanoencapsulation-92.4 ± 1.6[26]


### 3.3. Characterization of the Microcapsules

#### 3.3.1. Particle Size

Figure 3A shows the microphotograph of the oil droplets in the oil-in-water emulsion, and Figure 3B shows the dehydrated SFN microcapsules resuspended in an aqueous solution. The average diameter of the oil droplets before dehydration was 6.9 ± 2.3 µm, in the same order of magnitude as that reported by Saito et al. [30] for *Camellia* oil emulsion (equal to 3.9 µm). Zhu et al. [31] reported a 3.9 µm for *Perilla* seed and soybean oils emulsion microcapsules, agreeing with the values obtained in the present work. Hwangbo et al. [32] informed microcapsule diameters between 2 and 320 µm for ultrasonic emulsification using 1% olive oil, showing a high dispersion. The mean diameter obtained in resuspended dehydrated microcapsules in an aqueous solution (23.3 ± 18.1 µm) was considerably larger than the one reported by Wu et al. [10], who found a mean diameter of 2–4 µm for SFN microcapsules obtained via spray drying using maltodextrin as a material wall. This difference may be most likely due to a possible coalescence of the microcapsules as they come in contact with a hydrophilic environment after dehydration. The external wall material is hydrosoluble, and the inner layer of the microcapsules is hydrophobic. Accordingly, the system will move towards equilibrium and reduce entropy by reducing the hydrophobic surface in contact with the hydrophilic environment, resulting in microcapsules coalescence. In addition, the different drying methods used in this work (vacuum oven at 37 °C) and different wall materials should affect the particle size of the rehydrated SFN microcapsules.

Figure 4 shows SEM images obtained for the dehydrated SFN microcapsules. The microstructure analysis shows spherical microcapsules with a size distribution ranging from 1 to 20 µm. Some agglomerates of larger size, as well as some microcapsules with broken and/or melted walls, are also present. The average diameter of the microcapsules was 3.555 ± 1.667 µm, agreeing with the diameter informed by Wu et al. [10] for maltodextrin SFN microcapsules. The SFN microcapsules showed a homogeneous distribution of particle diameter, which would favor their application in the food industry since homogeneous size distribution is associated with good flow properties [12]. The SEM images agree with the results of optic microscopy (Figure 3). When the dry microcapsules were rehydrated, a wider range of particle size distribution was observed compared to the original microcapsules suspension before drying. This phenomenon normally happens when a microparticle system is oven-dried. At higher temperatures, the GA + Vaseline system loses viscosity; therefore, some microcapsules rupture in smaller ones, and some others coalesce in larger ones. 

#### 3.3.2. SFN Content in the Microcapsules

The SFN content in the microcapsules was assessed by RP-HPLC. Figure 5 shows the chromatograms obtained for (A) SFN standard, (B) pure microcapsules, (C) SFN extract, and (D) microencapsulated SFN. The SFN peak appears at a retention time of 2.5 min at A_254_. The SFN peak appears in the extract and in the SFN microcapsules, thus confirming that the compound was incorporated in the microcapsules. In Figure 5C,D, there are peaks at 1.9, 2.0, and 2.9 min, which probably correspond to compounds present in the broccoli seeds extract that were also incorporated in the microcapsules.

The SFN content in the microcapsules obtained at the optimum condition was 333.18 ± 18.1 [μg SFN/g dry powder]. This content is 2-fold higher than the reported for complex coacervation with gelatin–gum Arabic and 4-fold higher than that informed for gelatin–pectin coacervation [14]. The SFN content found in the present work is 3-fold higher than the one reported by Yepes-Molina et al. [25] in 2 mL of SFN encapsulated in broccoli membrane vesicle. Accordingly, the method developed in this work gives the highest SFN content in microcapsules reported so far. 

#### 3.3.3. Color

The SFN microcapsules were characterized in terms of color. Figure 6 shows images of the dehydrated SFN microcapsules, which exhibit a homogeneous white-yellow color. This is confirmed by the CIEL*a*b* values, with almost no traces of red or green (a* near 0.0) and slightly yellow traces with b* near 10, and a much more marked white color (luminosity) with L* near 90. 

The powder was homogeneous in particle size after drying, and the lumps were easy to disaggregate. The white color is characteristic of microcapsules with carbohydrate walls in general. The CIEL*a*b* parameters were coherent with what was observed with the naked eye since the values obtained corresponded to a white powder. L* defines lightness with high positive values corresponding to a solid, whiter color. The b* parameter correlates with the slight yellowness of the sample. Since those two parameters are relevant for white samples, the a* value is not normally taken into account to compare products with different degrees of whiteness. When compared with the literature, the L* and b* parameters are equivalent to those reported for goat cheese (87.1 ± 14.8 and 8.2 ± 4.4), cow and goat yogurt (86.2 ± 11.0 and 9.0 ± 5.3; 86.6 ± 6.0 and 12.1 ± 7.0). Cow and goat milk powders (88.6 ± 10.1 and 11.8 ± 6.1; 89.2 ± 6.0 and 11.0 ± 2.8) were also within the color range of the SFN microcapsules [33]. These L* and b* values are also equivalent to what was reported by Chudy et al. [34] for cow milk yogurt (92.7 and 9.14) and whey (95.46 and 13.93). In the study by Arepally and Goswami [35], L* values within 91.35 and 92.82 and b* values of 5.10 to 5.11 were determined for probiotic microcapsules spray dried at 150 to 130 °C and gum Arabic contents of 5 and 7.5%, respectively. These results show that SFN microcapsules will be suitable for use in milk and other dairy products.

#### 3.3.4. Fourier Transform Infrared Spectroscopy (FTIR) Analysis

Figure 7 shows the FTIR spectra obtained for SFN microcapsules and gum Arabic. The FTIR spectrum of SFN has been described in several publications. Normally, it shows bands at 2181, 2107, 1452, 1350, 1252, and 1025 cm^−1^. Absorbance at 2181 and 2107 cm^−1^ due to the –N=C=S stretching and absorbance peak at around 1025 and 1252 cm^−1^ due to the S=O, C-N bonds are the most characteristic of this molecule [10,36]. 

The FTIR spectrum of gum Arabic showed the characteristic bands for inter-OH and intra-OH stretching at 3000–3600 cm^−1,^ always present in carbohydrates. Peaks at 2980 and 2880 cm^−1^ are associated with -CH_2_ stretching, and bands at 1598 cm^−1^ and 1374 cm^−1^ are characteristic of -COO asymmetric and -COO- symmetric stretching vibrations, respectively. The bands at 963 cm^−1^ are associated with -CO stretching, and the absorption at 1028 cm^−1^ indicates C-O-C stretching. These bands are well in line with what is described by Stephanovic et al. [37] and Ai et al. [38].

Figure 7 shows all the characteristic and more differentiating bands of SFN: the peaks at 1452 cm^−1^, 1062 cm^−1^, and 704 cm^−1^ corresponding to the stretching vibration of C–S, C−N, and S=O bonds, respectively, and that only exist in this molecule. The FTIR analysis therefore confirms that SFN and gum Arabic are the main molecular components of the microcapsules obtained.

### 3.4. Thermal Stability of Free and Microencapsulated SFN 

The degradation kinetics of free and microencapsulated SFN in an aqueous solution was studied assuming a first-order kinetic model in the range of 37 °C to 70 °C, as shown in Figure 8. At all temperatures, free SFN exhibited faster degradation, as confirmed by the degradation kinetics constants, which were one order of magnitude higher than those obtained for microencapsulated SFN. This demonstrates that an oil-in-water emulsion system using gum Arabic as wall material increases the thermal stability of SFN significantly. 

Table 4 shows the degradation kinetic constants of free and microencapsulated SFN. The greatest difference between free and microencapsulated SFN was observed at 50 °C, where microencapsulation decreased by 6-fold in the degradation kinetic constant of SFN. The values obtained in the present work agree with other methods reported in the literature. At 50 °C, Fahey et al. [11] reported k = 0.004 [h^−1^] using α-cyclodextrin followed by lyophilization, and Wu et al. [29] obtained k = 0.002 [h^−1^] using HP-β-CD inclusion complex, which was in the same order of magnitude of the constant reported here (0.0081 h^−1^). However, the oil-in-water emulsion system allows significantly higher entrapment efficiencies, making it more attractive for potential industrial use. A similar situation occurs at 60 °C, agreeing with the degradation constants reported by Wu et al. [12] using maltodextrin and gum Arabic followed by spray drying. The degradation constant obtained for microencapsulated SFN at 70 °C was 8-fold lower than that reported by Wu et al. [12] at 75 °C using HP-β-CD inclusion complex (k = 0.250 h^−1^) at neutral pH. Accordingly, the microencapsulation method presented in this work has a considerably higher stabilizing effect on SFN at temperatures above 60 °C than the methods reported so far.

The activation energy (Ea) was estimated for the free and microencapsulated SFN considering data at 50, 60, and 70 °C (Table 4) since SFN begins its degradation at 40 °C [24], and therefore at temperatures below 40 °C the Arrhenius model does not explain the observed behavior. Microencapsulated SFN showed a higher Ea than free SFN, confirming the thermal protective effect of this microencapsulation method.

The Ea values obtained here are in the same order of magnitude as those reported by Mahn et al. [24] for SFN degradation during broccoli dehydration in a tray dryer, ranging between 58 and 70.4 KJ/mol. Wu et al. [10] reported Ea between 70.7 and 90.6 KJ/mol for free SFN at different conditions of pH (2–6) and temperatures (60–90 °C). Van Eylen et al. [39] obtained an Ea of 89.03 KJ/mol for SFN in broccoli juice.

## 4. Conclusions

The O/W microencapsulation method was effective in increasing SFN stability in aqueous solutions, resulting in degradation kinetics constants found in the same order of magnitude as the reported for other methods at 50 and 60 °C. This method gave the lowest degradation constant at 70 °C in comparison with those reported in the literature, suggesting a higher stabilizing effect at high temperatures in comparison with the currently available methods. The optimization of the microencapsulation process allowed for reaching an entrapment efficiency of 65%, which is the highest reported for any SFN microencapsulation method so far. These results are expected to contribute both to the more efficient incorporation of the beneficial properties of SFN in various food matrices (liquid beverages, dairy products, solid formulations, and food supplements) and to the exploration of new encapsulation technologies that can maximize efficiency and stability of the SFN.

## Figures and Tables

**Figure 1 foods-12-03869-f001:**
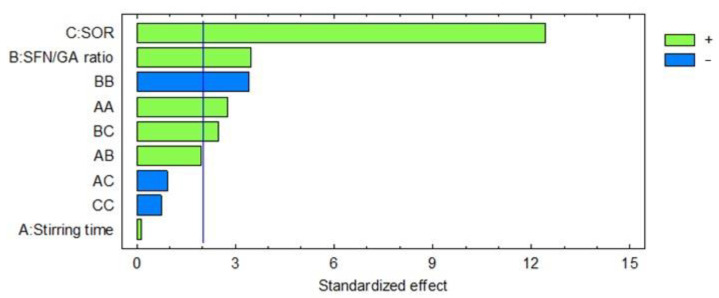
Pareto chart analysis of the Box–Behnken design to maximize SFN entrapment efficiency. The vertical line indicates significance at a 95% confidence level.

**Figure 2 foods-12-03869-f002:**
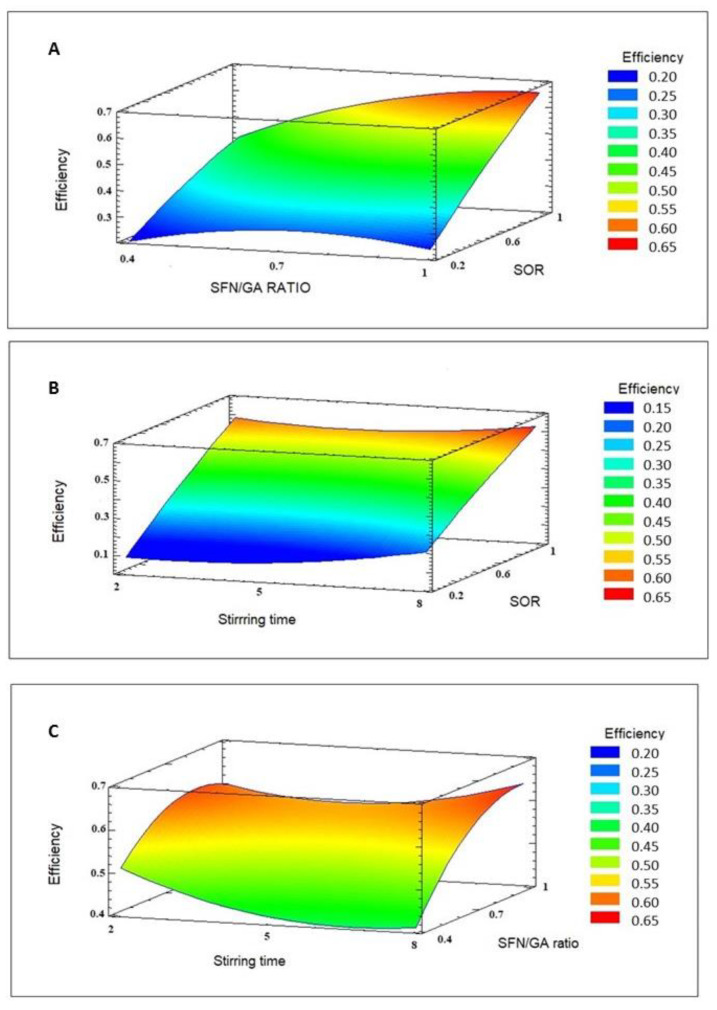
Response surface obtained for entrapment efficiency. (**A**) SFN/GA ratio vs. SOR for stirring time: 8 min; (**B**) stirring time vs. SOR for SFN/GA ratio: 1.0; (**C**) stirring time vs. SFN/GA ratio for SOR: 1.0.

**Figure 3 foods-12-03869-f003:**
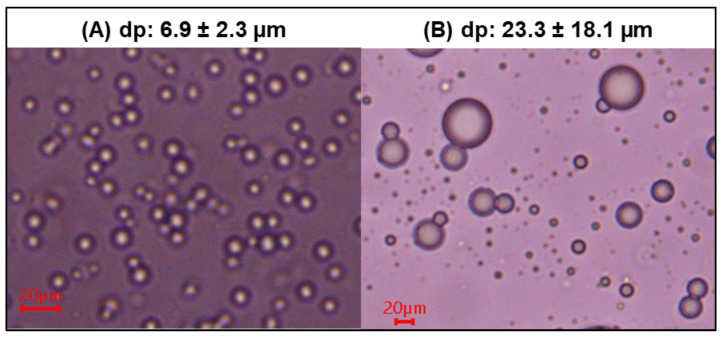
Optical microscopic photograph. (**A**) Oil droplets in O/W emulsion (objective 100×); (**B**) dehydrated SFN-microcapsules resuspended in water (objective 40×); dp = particle diameter.

**Figure 4 foods-12-03869-f004:**
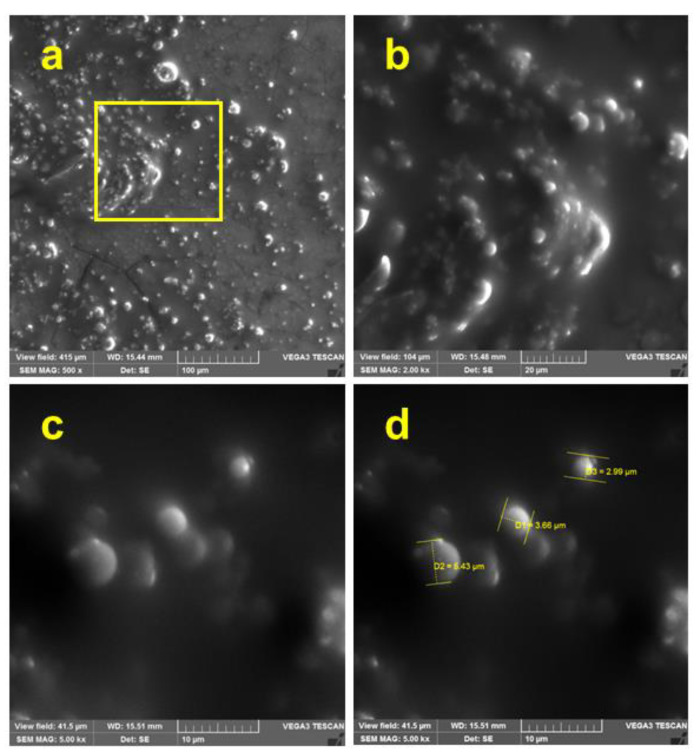
Scanning Electron Microscopy (SEM) of SFN microcapsules. (**a**) General view of a group of microcapsules where the size distribution of can be observed; (**b**) a close-up view of the yellow square area highlighted in photomicrography (**a**); (**c**) a magnified view of individual microcapsules showing the regular spherical shape of most of the microcapsules obtained; (**d**) the diameter measure of three of the microcapsules shown in photomicrography (**c**).

**Figure 5 foods-12-03869-f005:**
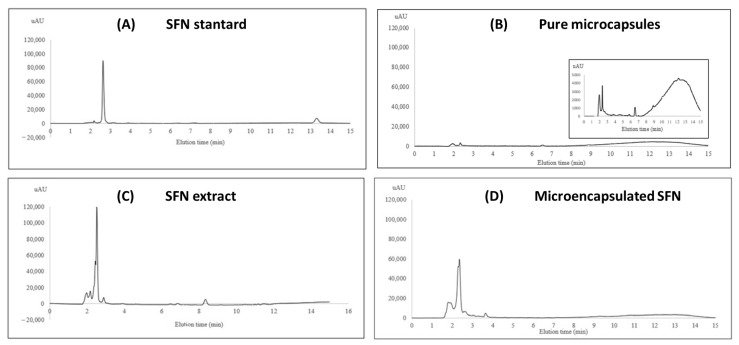
RP-HPLC chromatograms of (**A**) SFN standard, (**B**) pure microcapsules, (**C**) SFN extract, and (**D**) microencapsulated SFN.

**Figure 6 foods-12-03869-f006:**
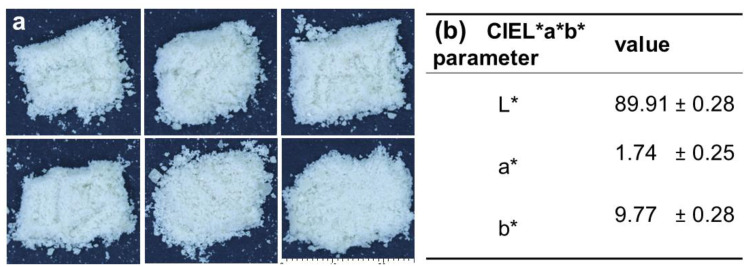
(**a**) Photograph of six replicate samples of dry powder of SFN microcapsules, (**b**) CIEL*a*b* color parameters for the SFN microcapsules in CIEL*a*b* color space.

**Figure 7 foods-12-03869-f007:**
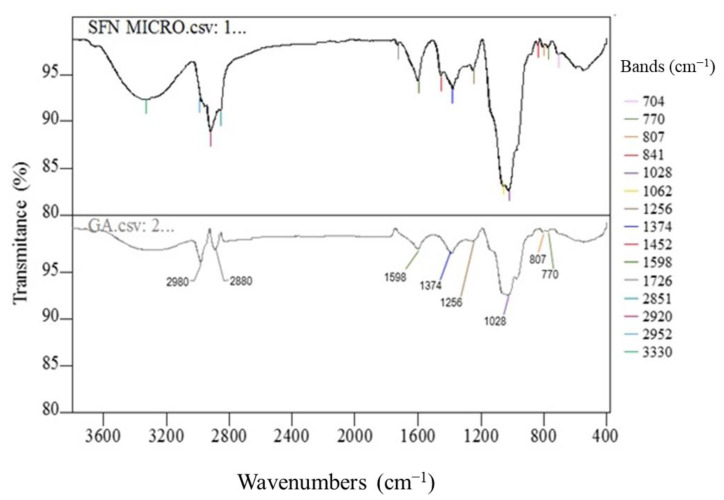
FTIR spectra of sulforaphane microcapsules (SFN MICRO) and gum Arabic (GA) showing lines of different colors for each band identified.

**Figure 8 foods-12-03869-f008:**
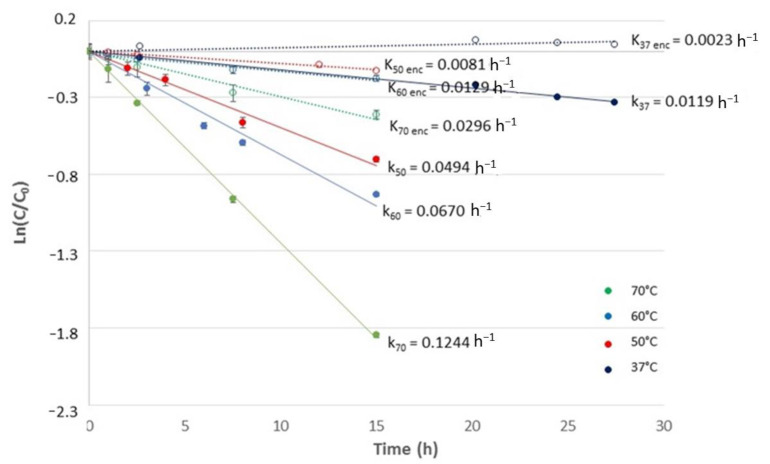
Estimation of degradation kinetics constants for encapsulated SFN (dotted lines and open circles) and free SFN (solid lines and filled circles) at 37, 50, 60, and 70 °C. Individual points represent the experimental data, and the lines correspond to the linear regression model adjusted to the data.

**Table 1 foods-12-03869-t001:** Experimental design (multilevel factorial 2^1^ × 4^1^) to determine the conditions that maximize the physical stability of the emulsions. Volume: 10 mL; stirring time: 5 min; stirring rate: 8000 rpm; oil phase: Vaseline; wall material: gum Arabic; SOR: 1/6 (ml/mL); surfactant: Tween 80. The selected condition is highlighted in bold.

Sample	Oil Phase/Aqueous Phase (mL/mL)	Gum Arabic Concentration (% w/w)	Time Before Creaming Starts (min)	Time for Complete Phase Separation (min)
A1	1/6	20	23 ± 2	56 ± 4
A2	1/4	20	22 ± 2	56 ± 2
A3	1/2	20	69 ± 3	246 ± 5
A4	1/1	20	248 ± 6	300<
A5	1/1	10	43 ± 2	93 ± 7
A6	1/2	10	22 ± 1	68 ± 5
A7	1/4	10	2 ± 0.2	5 ± 0.1
A8	1/6	10	1 ± 0.2	3 ± 0.1

**Table 2 foods-12-03869-t002:** Optimization of the microencapsulation conditions. SFN/GA = SFN/gum Arabic mass ratio; SOR = surfactant to oil phase mass ratio; EE = entrapment efficiency. Stirring rate = 8000 rpm, total volume = 10 mL, oil phase = Vaseline: ethanol 50% (v/v), gum Arabic concentration = 20% (w/v). The selected condition is highlighted in bold.

Run	Stirring Time (min)	SFN/GA (µg SFN/mg GA)	SOR (g Tween 80/g Oil Phase)	EE (%)
1	5	0.4	1.0	34 ± 3
2	2	0.7	1.0	63 ± 3
**3**	**8**	**0.7**	**1.0**	**65 ± 4**
4	2	0.4	0.6	43 ± 3
5	5	1.0	1.0	53 ± 3
6	8	1.0	0.6	43 ± 1
7	5	0.4	0.2	23 ± 1
8	8	0.4	0.6	34 ± 4
9	8	0.7	0.2	29 ± 0
10	2	1.0	0.6	41 ± 1
11	5	1.0	0.2	28 ± 1
12	2	0.7	0.2	22 ± 2
13	5	0.7	0.6	41 ± 3
14	5	0.7	0.6	42 ± 1
15	5	0.7	0.6	41 ± 0

**Table 4 foods-12-03869-t004:** Kinetic constant (k), activation energy (Ea), regression coefficient (R^2^), and frequency factor (k_0_) for degradation of free and microencapsulated SFN in aqueous solution.

	Free SFN				Microencapsulated SFN			
T [°C]	k [h^−1^]	Ea [KJ/mol]	R^2^	k_0_ [h^−1^]	k [h^−1^]	Ea [KJ/mol]	R^2^	k_0_ [h^−1^]
70	0.1244	42.3	0.96	2.8 × 10^4^	0.0296	59.6	0.97	9.4 × 10^8^
60	0.0670				0.0129			
50	0.0495				0.0081			
37	0.0119				0.0023			

## Data Availability

The data presented in this study are available on request from the corresponding author.

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
