# Peer review of "Optimization of a Microencapsulation Process Using Oil-in-Water (O/W) Emulsion to Increase Thermal Stability of Sulforaphane"

_foods, 2023, doi:10.3390/foods12203869_

Round 1

Reviewer 1 Report

I reviewed the manuscript titled “Optimization of a microencapsulation process using oil-in-water (O/W) emulsion to increase thermal stability of Sulforaphane. The manuscript is well written and contributes to the field. However, characterization of microcapsules must be provided with few more experiments like color, FTIR and SEM etc

Abstract

Conclusions and recommendations must be added in abstract

Introduction

Lines 93-98: objectives should be addressed clearly

Methodology

Line 109: method proposed in [22]….. should be revised as method proposed by X et al. [22]

Provide citation for section 2.4. Microencapsulation of SFN

Equation 1: what is the unit %? Or g? What is the unit of SFN content in the microcapsules and SFN content in the extract?

What is the need of performing thermal stabilities studies at higher temperatures like 50, 60, and 70? Authors performed at 37 and suddenly jumped to > 50 °C, why?

2.5. Characterization of microcapsules: authors studied only two parameters. I suggest to include color, FTIR, SEM

Encapsulation efficiency is less. The max is 65%. This indicates less encapsulation efficiency. There is no good quality wall material, which might have serious effect during storage studies

Figures 4 and5: quality must be improved

Authors claim that the microcapsules degrade at 40 °C and what is the need of performing stability at higher temperature. I believe no industry stores microcapsules at 50/60/70 °C. What would be the practical application of studying thermal stability at 50/60/70 °C.

3.3.2. SFN content in the microcapsules: this section should be discussed and compared with literature thoroughly

3.3.1. Particle size: this section must be discussed thoroughly and compare with available literature

Methodology on activation energy must be provided in methodology section

Provide the graphical abstract for easy understanding

References must be cross-checked and must align with journal format

Scientific names must be in Italics. Please revise all scientific names 

Reviewer 2 Report

1. This o/w system is designed for use in foods; why is vaseline rather than edible oil used as an oil phase to dissolve SFN? (line#134)

2. Where is the standard sulforaphane compound obtained from?

3. What does “350 measurements” mean? What are the criterions of sampling the emulsion and selecting viewing areas? How many particles are counted in one measurement? (line#151)

4. SFN has poor water solubility. How can it be taken evenly for analysis when the free SFN extract was suspended in water in the control group? (line#179)

5. How do you define and correctly determine “First phase separation” time and “Total phase separation” time?

Round 2

Reviewer 2 Report

The questions have been answered.